# Influence of Sucrose and Activated Charcoal on Phytochemistry and Vegetative Growth in *Zephyranthes irwiniana* (Ravenna) Nic. García (Amaryllidaceae)

**DOI:** 10.3390/plants13050569

**Published:** 2024-02-20

**Authors:** Bertholdo Dewes Neto, Kicia Karinne Pereira Gomes-Copeland, Dâmaris Silveira, Sueli Maria Gomes, Julia Marina Muller Craesmeyer, Daniela Aparecida de Castro Nizio, Christopher William Fagg

**Affiliations:** 1Department of Botany, University of Brasília, Brasília 70910-900, DF, Brazil; bertholdo.dewes@outlook.com (B.D.N.); suelimariagomes@gmail.com (S.M.G.); 2Laboratory of Natural Products, Faculty of Health Sciences, University of Brasilia, Brasília 70910-900, DF, Brazil; kiciagomes@gmail.com (K.K.P.G.-C.); damaris@unb.br (D.S.); juliamm@unb.br (J.M.M.C.); 3Department of Agronomy, University of Sergipe, São Cristóvão 49107-230, SE, Brazil; danielanizio@yahoo.com.br

**Keywords:** lycorine, histolocalization test, vegetative growth, sucrose, activated charcoal, phytochemistry

## Abstract

*Zephyranthes irwiniana* (Ravenna) Nic. García is an endemic, red list threatened species from the Brazilian savanna (Cerrado) with pharmacological potential to treat the symptoms of Alzheimer’s Disease (AD). This work analyzed the vegetative growth and phytochemistry of its potential compounds, in response to variations in sucrose concentration and activated carbon (AC). Seeds were germinated *in vitro* and in the greenhouse. The *in vitro* bulbs were separated in six treatments with different sucrose concentrations (30, 45 and 60 gL^−1^) and/or AC (1 gL^−1^). Biomass increases in individuals grown in the greenhouse were higher than those cultivated *in vitro*. Sucrose concentration significantly increased biomass and root number. AC had a positive influence on leaf and root size, and a negative influence on root number. GC–MS analyses indicated great variation in the abundance of α-terpenyl-acetate, ethyl linoleate, clionasterol and lycorine between treatments, with maximum concentrations of 53.06%, 38.68, 14.34% and 2.57%, respectively. Histolocalization tests indicated the presence of alkaloids in the leaf chlorenchyma and bulb cataphylls. Finally, the present study provided new evidence that the constitution of the culture medium directly influences the vegetative growth and phytochemistry of this species, providing a good medium condition for propagating the species under threat.

## 1. Introduction

The increased demand of natural products has stimulated, in recent decades, the development of techniques for the mass production of plants [1,2], and the importance of *in vitro* propagation as a biotechnological tool [3], which also enables the production of bioactive compounds from plants under stable and controlled conditions in a short period of time [2].

The Amaryllidaceae family is known for its ornamental potential and the ability to synthesize several alkaloids, including galantamine and lycorine, and potent acetylcholinesterase (AChE) inhibitors, used to alleviate the symptoms of Alzheimer’s Disease (AD) [4]. However, some species in this family are poorly represented, have a low multiplication rate [5] and are vulnerable or at risk of extinction [6]. Among the species in this family, *Zephyranthes irwiniana* (Ravenna) Nic. García (Figure 1) is endemic to the Brazilian savanna (Cerrado), classified as Vulnerable in terms of extinction threat [7]. Few studies have evaluated the biological properties of this species [8], but there is no record of its traditional and medicinal use in Brazil and no studies on its propagation or phytochemistry under controlled conditions.

Recent studies with Amaryllidaceae species have shown that propagation methods [9] and the plant habitat [10] can alter phytochemistry, varying the abundance of secondary metabolites for the same species. Many *in vitro* propagation studies evaluated the influence of culture medium composition on the seedling phytochemistry and opted to vary the type and concentration of the carbohydrate source, with sucrose being the most common [11,12]. According to the literature [13], increasing the concentration of sucrose in the culture medium up to 90 gL^−1^ increases the biomass of some Amaryllidaceae species. However, few have evaluated phytochemistry from seed propagation [9] and the interaction of sucrose with activated charcoal on vegetative growth and phytochemistry. We expect that in addition to effects of sucrose, the activated charcoal can modify the biomass increase rate and the phytochemistry due to its adsorption capacity [14].

These studies are important to increase biomass production, improve yields and the relative abundance of target compounds [12]. Furthermore, evaluating vegetative growth and determining the localization of these compounds in plant tissues can provide a better understanding of the role and sites of synthesis and accumulation of these compounds [15]. However, the data of histolocalization of alkaloids in Amaryllidaceae are scarce.

Given this set of factors, the objective of this work was to analyze, for the first time, the vegetative growth and phytochemistry of potential compounds for the treatment of AD, in response to variations in sucrose concentration and the use of activated charcoal in the culture medium, for the species *Z. irwiniana* propagated *in vitro* by seed.

## 2. Results

### 2.1. Vegetative Growth in a Greenhouse

The beginning of germination in a greenhouse occurred 20 days after sowing. Only 11 seeds germinated out of a total of 162, representing 6.79% of the total. The average maximum and minimum temperatures recorded during the experiment were 26.66 °C and 12.91 °C, respectively. The maximum, average and minimum temperatures recorded were, respectively, 44.91 °C, 20.03 °C and 6.13 °C. The maximum, average and minimum relative humidities recorded were, respectively, 99%, 67% and 17%.

Due to the low germination rate, only seven individuals were evaluated. The leaf number and average length were 1.95 and 9.38 cm, respectively, and for the root, 4.86 and 14.73 cm. The total fresh and dry biomass was 45.82 g and 7.41 g, and the average was 6.54 g and 1.05 g, respectively. The dry biomass of all leaves, roots and bulbs was, respectively, 0.53 g, 1.04 g and 5.84 g.

### 2.2. Vegetative Growth In Vitro and Influence of Sucrose and Activated Charcoal

The first seeds germinated after the fifth day of sowing, and in total, 154 seeds germinated, representing 96.86% of germinability.

Table 1, Table 2 and Table 3 show the results on the sucrose and activated charcoal influences on the vegetative growth *in vitro* of *Z. irwiniana*.

After 5 months of *in vitro* cultivation, it was observed that the increase in sucrose concentration increased the dry biomass per individual without a significant interaction of activated charcoal (sucrose biomass: 30 gL^−1^, 0.05 g; 45 gL^−1^, 0.06 g; 60 gL^−1^, 0.10 g). Furthermore, it increased in fresh biomass per individual (sucrose biomass: 30 gL^−1^, 0.44 g; 45 gL^−1^, 0.66 g; 60 gL^−1^, 0.83 g) and the number of roots (sucrose roots number: 30 gL^−1^, 5.33; 45 gL^−1^, 7.97; 60 gL^−1^, 10.89) (Table 1). For fresh biomass and number of roots, there was a significant interaction of activated charcoal only at a concentration of 60 gL^−1^ of sucrose (Table 3).

The activated charcoal (AC) significantly increased the length of the largest leaf (with AC: 21.76 cm; without AC: 18.31 cm) and the largest root (with AC: 20.16 cm; without AC: 14.95 cm) (Table 1) at all levels of sucrose concentration, without the significant interaction of sucrose within AC levels. The AC also positively influenced the number of leaves, with a significant interaction for the sucrose concentration of 45.0 gL^−1^ (T3: 3.61; T4: 4.11) and 60.0 gL^−1^ (T5: 3.61; T6: 4.44) (Table 3), however, there was no significant interaction of sucrose within the AC levels. The number of roots was negatively influenced by the AC (with AC: 7.24 cm; without AC: 8.89 cm) (Table 1), interacting significantly with sucrose only at a concentration of 60 gL^−1^ (T5: 12.22; T6: 9.56) (Table 3). There was no influence of AC on the average fresh and dry biomass per individual (Table 1, Table 2 and Table 3). The total fresh biomass of each treatment was 10.52 g (T1), 8.48 g (T2), 14.50 g (T3), 13.90 g (T4), 13.42 g (T5) and 18.30 g (T6), and dry biomass was 0.88 g (T1), 1.06 g (T2), 1.39 g (T3), 1.15 g (T4), 1.72 g (T5) and 2.08 g (T6).

### 2.3. Ethanolic Extract

The treatment GH presented the lowest percentage yields of compounds extracted using ethanol for the bulb (1.85%), leaf (13.36%) and root (6.94%) (Figure 2). The highest yields obtained for ethanolic leaf extract were from treatments without AC and the treatment T3 (46.71%) was significantly higher than all other treatments. Except for treatments T3 and GH, the other treatments did not differ statistically. Bulb extract production showed similar results for all treatments, except for T6 (4.42%) and GH (1.85%), which had significantly lower values. The results for the bulbs, when compared with those obtained for the leaves, showed a significantly lower yield, however, the production of ethanolic extract for the root was similar for both.

Treatments with a lower concentration of sucrose per liter showed a better percentage extraction yield, observing a downward trend in treatments with more than 45 gL^−1^ of sucrose. The best results obtained were from treatments T3 and T2, respectively, for leaf and root. For the root, there was a significant interaction between activated charcoal and a concentration of 30 gL^−1^ of sucrose, and for the leaf, activated charcoal interacted with sucrose, negatively influencing the T4 extraction yield. The sucrose concentration did not significantly influence the bulb extraction yield; this only occurred when it interacted with AC in the T6 treatment.

### 2.4. Phytochemical Profile GC–MS

The characterization of the chemical profile of all extracts revealed 15 potential compounds (Table 4). For analysis, the compounds α-terpinyl-acetate, ethyl linoleate, clionasterol and lycorine were selected, as they were detected in at least 80% of the extracts, except for lycorine, because it is a potential alkaloid for the AD treatment. Most importantly, they have characteristics of medicinal interest according to the scientific literature [16,17,18,19]. The retention time and variation in absolute intensity of each treatment are described in the chromatogram in Figure 3 and Figure 4. All chromatograms are supplied in Appendix A.

#### 2.4.1. Alpha-Terpinyl-Acetate

The T2 ethanolic extract (30 gL^−1^ sucrose and 1.0 g AC) presented the highest percentage of area in the α-terpinyl-acetate chromatogram, being 10.70% and 53.06%, respectively, for bulb and leaf; however, the root extract showed no difference between treatments. The results of this study demonstrate that the abundance of α-terpinyl-acetate varied due to the treatment with sucrose and AC, with a positive interaction between both in the T2 treatment for leaf and bulb, and negative in T3 for the root, as the extract without AC presented a value higher than the extract with AC (T4). It was observed that the abundance of α-terpinyl-acetate decreased from the leaf to the bulb and from the bulb to the root in T2, and the opposite occurred with T3, where there was an increase in abundance in this direction. The GH treatment (substrate in a greenhouse) did not present an advantage in production when compared to the *in vitro* treatment, hence *in vitro* cultivation is a promising option to increase the production of this compound.

#### 2.4.2. Ethyl Linoleate

The EE from the treatment GH (substrate in a greenhouse) presented the highest percentage of area for the bulb and root, respectively, at 38.68% and 38.41%, compared to the other treatments (T1-T6), and these did not show a significant difference between them. The compound was not identified in the leaf and root extract of T2 (sucrose 30 gL^−1^ and 1.0 g of AC) and in the leaf extract of T3 (sucrose 45 gL^−1^ without AC). The leaf extracts presented an abundance that varied between 1.59% and 6.98%, well below the maximum abundances found for the bulb and root. This suggests that the leaf is not promising for the extraction of this compound compared to the bulb and root. For the other extracts, there was no significant difference when analyzed and compared within the same part of the plant.

#### 2.4.3. Clionasterol

Root extracts presented the two highest relative abundances of 14.34% and 13.94%, respectively, for T6 and T4, being significantly higher than the other extracts. Treatments T2 and T5 presented the lowest abundances, and both were within the confidence interval. The other treatments were statistically equal. Bulb treatments T2–T5 presented relative abundance between 1.00% and 2.00%, and significantly higher values were obtained in the treatments T1 (5.60%) and T6 (4.58%). In general, the T6 treatment presented the best results, as for the bulb and root it was statistically superior to the others; for the leaf, it presented the second highest abundance. The GH treatment presented the highest abundance of ethanolic extracts in the leaves, at 11.62%, and the lowest abundance in the roots (0.73%). Interestingly, treatment T1 also showed good results in general terms, similar to T6; however, the composition of the culture media differs in the concentration of sucrose and the use of AC. T6 was higher only in the root extract and T1 in the leaf and bulb extract.

#### 2.4.4. Lycorine

The alkaloid lycorine was identified in the bulb of treatments T1, T3, T6 and GH, presenting, respectively, a relative abundance of 0.74%, 0.32%, 1.09% and 0.59%, and in the leaf of treatments T1, T5, T6 and GH, respectively, 0.87%, 2.57%, 0.86%, 0.54%. No alkaloids were detected in root extracts. Treatments T1, T6 and GH presented values close to relative abundance for the bulb and leaf; however, for the leaf, T5, when compared to T6, presented a much higher value indicating that AC interacted negatively with the sucrose concentration of 60 gL^−1^. The same occurred at a concentration of 30 gL^−1^ of sucrose, as lycorine was not detected in T2 in any part of the plant. Furthermore, the treatment that presented the highest relative abundance was the T5 leaf extract. For the bulb, AC interacted positively with the concentration of 60 gL^−1^ of sucrose, as lycorine was detected in T6 but not in T5. The opposite happened at concentrations of 30 and 45 gL^−1^. The best sucrose concentration was 60 gL^−1^ for the bulb and leaf presenting, respectively, 1.09% (T6) and 2.57% (T5) of relative abundance.

### 2.5. Histolocalization of Alkaloids in Leaves and Bulblets

The histochemical tests are shown in Figure 5 and the intensity of reaction in Table 5. The histochemical test on individuals from T1 treatment (*in vitro*) with Dragendorff’s and Wagner’s reagents were positive for leaf and bulb, where the formation of a brownish-red color was observed in both, indicating the presence of alkaloids and dark, brown-colored starch grains. In the leaves, alkaloids were detected only in the palisade and spongy chlorenchyma cells, present in several cells in the leaf margins just below the adaxial epidermis. Alkaloids were detected in the chlorenchyma cells that accompany the adaxial and abaxial epidermis. No alkaloids were detected in the spongy parenchyma, vascular bundles, or other structures present in the leaf.

In the bulb, the reaction for alkaloids was stronger in the region adjacent to the abaxial than the adaxial epidermis of each cataphyll. The internal mature cataphylls have more mesophyll layers with strong reaction than those external. It was observed that the concentration of starch grains is organized in a similar way to the alkaloids in the cataphyll; that is, regions of the cataphyll with the highest concentration of alkaloids are the same ones that present the highest concentrations of starch grains.

## 3. Discussion

The present study showed that the concentration of sucrose and the use of AC in the culture medium significantly influences the vegetative growth and, mainly, the phytochemistry of *Z. irwiniana*. Individuals propagated in a greenhouse had a low germination rate (6.79%), much lower than that obtained using *in vitro* cultivation (96.86%). The results obtained using *in vitro* propagation were very promising, being an efficient strategy for propagating the species *Z. irwiniana*, and the low germination rate recorded in the greenhouse propagation agrees with Herranz et al. (2020), who state that the Amaryllidaceae family has a small natural multiplication rate [20]. The germination success obtained through *in vitro* propagation was higher than that observed for other Amaryllidaceae species. [21,22,23].

The increase in the biomass of individuals grown in a greenhouse (GH 6.54 g) was higher than that *in vitro* (T1: 0.47 g; T2: 0.40 g; T3: 0.64 g; T4: 0.67 g; T5: 0.71 g; T6: 0.96 g), with an emphasis on the bulb which represented, on average, 78.8% of the total dry biomass (root, leaf and bulb). The total fresh and dry biomass, obtained by the sum of individuals grown in a greenhouse, was much higher when compared to those obtained in all *in vitro* cultivation treatments (GH: 45.82 g; T1–T6: 10.52 g–18.30 g), regardless of the sucrose concentration and the use of AC in the culture medium. These results are even more significant when we compare the number of individuals used to obtain the total biomass per treatment, which was 7 individuals in greenhouse cultivation and an average of 21 individuals for each treatment cultivated *in vitro*. Therefore, the present study suggests that cultivation in a greenhouse is more advantageous in terms of increasing biomass than *in vitro*, especially if the target is the bulb.

The best biomass productivity observed *in vitro* was obtained using a concentration of 60 gL^−1^ of sucrose, which also positively influenced the number of roots (Table 1). Furthermore, this concentration did not negatively influence the other variables. Juan-Vicedo et al. (2019) observed an increase in biomass production of the species *Lapiedra martinezii* Lag. (Amaryllidaceae) related to the highest concentration of sucrose, and the greatest increase in biomass production found was also for a concentration of 60 gL^−1^ of sucrose without activated carbon [11]. Other studies also observed that a higher concentration of sucrose in the culture medium improved bulb development in Amaryllidaceae cultivars and species [24,25,26]. However, these studies only evaluated biomass production related to variation in sucrose concentration, but not the interaction with AC. The interaction of sucrose at a concentration of 60 gL^−1^ with AC was positive to produce dry biomass and negative for the number of roots. It is interesting to highlight this, because the extraction of secondary metabolites is carried out from dry biomass and the use of AC can be a better alternative in the propagation of Amaryllidaceae species to produce secondary metabolites.

Other studies observed that the best concentration of sucrose in the culture medium for increasing biomass was 90 gL^−1^ [13,27]. This concentration was not tested in this study, and it would be interesting to test the interaction of AC with the sucrose concentration of 90 gL^−1^ in subsequent studies.

Different from sucrose, AC did not influence the increase in fresh and dry biomass, however, it had a significant and positive influence on the number of leaves and the length of leaves and roots without the significant interaction of sucrose concentrations. A few previous studies have evaluated the influence of AC on the *in vitro* vegetative growth of Amaryllidaceae, and the results obtained in the present study show that the use of AC can positively influence vegetative growth and phytochemistry; therefore, it is important to evaluate its effect on other species. Tahchy et al. (2011) observed that the addition of AC to the *in vitro* tissue cultures of three species of Amaryllidaceae was not beneficial at any concentration used (0.0, 5.0 and 10.0 gL^−1^), always having a negative effect on survival, callus and organogenesis. Furthermore, they observed that the concentration of AC in the nutrient medium decreases the rate of bulb and root formation [28]. Juan-Vicedo et al. (2019) did not observe differences in the morphometric characteristics between the seedlings of the species *Lapiedra martinezii* cultivated *in vitro* with AC and wild specimens, finding the stability of the materials produced *in vitro* [11]. In contrast with these studies, the present study observed that the use of AC resulted in the production of leaves and roots with greater length and, in the case of the number of leaves, stimulated production. Thus, the use of AC must depend on the proposed objectives, as it was positive for the length of leaves and roots, but negative for the number of roots. Anatomical studies may provide new information about the influence of sucrose and AC on the morphology of the species *Z. irwiniana* micro-propagated *in vitro*.

The plants propagated in a greenhouse had the greatest biomass increase when compared to seedlings propagated *in vitro* but had the lowest EE yields. This suggests that biomass productivity did not result in a greater productivity of metabolites, yet treatments with a lower concentration of sucrose per liter showed a better percentage extraction yield, observing a downward trend in cultures with more than 45 gL^−1^ of sucrose. When the objective is the production of secondary metabolites, improving the extraction yield is very important to improve the efficiency of the entire process, and high yields result in a lower cost of producing plant biomass, as fewer individuals are used to obtain the same quantity of extract. In this aspect, *in vitro* cultivation proved to be the best alternative for the production of crude EE, with an emphasis on the leaf, which presented the best extraction yields in crops that did not use AC. These results are possibly related to AC’s high capacity for adsorption of plant growth regulators and other organic compounds [29], as it is made up of a very fine network of pores and an extraordinarily large surface area [30].

Previous studies have investigated the phytochemical profile of plant species [31,32,33] or the variation in this profile and the relative abundance of the target compound for the same species, varying the method of propagation and extraction, and the composition of the culture medium or plant part used [9,13,27,33]. In contrast with these, the present study used AC to evaluate whether the interaction with sucrose at different concentrations would result in a variation in relative abundances for the same target compound. The variations observed in the chemical profile of the chromatograms of the crude EE (Figure 3 and Figure 4) in the present study revealed the influence of sucrose and AC on the biosynthesis of metabolites (Figure 3 and Figure 4), showing that they affected the relative abundance of the compounds α-terpenyl-acetate, ethyl linoleate, clionasterol and lycorine, in the leaf, bulb and root.

The α-terpenyl-acetate is a monoterpene ester [34]. This compound was studied for anticholinesterase activity and antioxidant activity; in addition, molecular anchoring, physicochemical and ADMET (absorption, distribution, metabolism, excretion and toxicity) properties were also determined to predict whether it is orally active and has properties central nervous system (CNS) medications [16]. Recently, studies on the inhibition of the SARS-CoV-2 virus were conducted [35].

Regarding the anticholinesterase potential of α-terpenyl-acetate, Chowdhury and Kumar (2020) concluded in their study that this compound binds to multiple drug targets implicated in AD and has also been demonstrated to be anticholinesterase, antioxidant, anti-amyloidogenic and hold neuroprotective potential, and could be used as a clue to develop a new, safe and effective therapy for AD [16].

Vaičiulytė et al. (2021) found a maximum relative abundance of 64.22% for the aerial parts of the species *Thymus pulegioides* L. [34] and in the study by Michet et al. (2008) a variation between 64.8 and 88.0% was reported, also for the aerial parts, with α-terpenyl-acetate being the dominant compound of the essential oil for this wild species [36]. Alam et al. (2019) found a relative abundance of between 41.42 and 55.36% for three varieties of *E. cardamomum* fruits [37]. For the species *Z. irwiniana*, a maximum relative abundance of 53.06% was recorded in the leaf extract of individuals cultivated at a concentration of 30 gL^−1^ of sucrose and 1.0 gL^−1^ of AC. This result suggests that the interaction of AC at this concentration of sucrose is very advantageous to produce α-terpenyl-acetate, as it was much higher than the relative abundance obtained in other extracts with the potential to replicate in *in vitro* cultures of different species that aim for the production of this compound.

Ethyl linoleate is an unsaturated fatty acid [38]. This compound has many physiological functions, such as increasing immunity, reducing cholesterol and lipid levels in the blood, and is the raw material for a highly effective medicine in the prevention and treatment of chronic diseases, such as cerebral thrombosis and atherosclerosis [39]. Kissling et al. (2005), using EE from *Crinum x powellii* bulbs, identified ethyl linoleate as responsible for the inhibition of acetylcholinesterase [17]. Furthermore, ethyl linoleate has antibacterial and anti-inflammatory properties [40].

Previous studies found an abundance of 15.86% in EE of *Phellinus linteus* (fungus) [41] and 19.67% for hexane extracts of crude oils of *Scutellaria edelbergii* Rench. f. [42]. Aly et al. (2022) found much lower values (0.83%) for hexane extracts from *Psidium guajava* L. leaves [18]. These results are lower than those obtained in the present study, which were 38.68% and 38.41%, respectively, for the bulb and root in the crude EE of individuals propagated in a greenhouse. Different to α-terpenyl-acetate, ethyl linoleate was more abundant in individuals propagated in a greenhouse. Furthermore, the influence of sucrose concentration and the use of AC in changing the abundance of this compound was not verified in this study, as they presented statistically equal values for the bulb and root, but not leaf extracts, which did not prove to be viable for the isolation and commercialization of the compound as they presented low abundance, varying between 1.59% and 6.98% in treatments.

Clionasterol (γ-sitosterol) has multiple bioactivities, such as anti-inflammatory, diabetes control [43], antitumor agent [44], antioxidant [18], bactericidal and fungicide [45]. Furthermore, recent studies report that clionasterol is an important agent in the enzymatic inhibition of acetylcholinesterase [18,46] and butyrylcholinesterase [47], presenting the best results in molecular coupling when compared to other compounds, indicating that it is a potential medicine for the treatment of AD.

*Zephyranthes irwiniana* showed promising results for clionasterol production, with a relative abundance of 14.34% in the root extract of individuals cultivated *in vitro* using 30 gL^−1^ of sucrose with AC. Aly et al. (2022) found a relative abundance of 3.90% in the hexanic extract of the species *Psidium guajava* L. [18], being lower than the maximum value recorded for *Z. irwiniana*. Other studies showed different clionasterol concentrations in plant extracts from different species, from 5.42% for the seed ethanolic extract of *Caesalpinia bonduc* (L.) Roxb. [47], 12.54% for the root methanolic extract of *Leptadenia reticulata* (Retz.) Wight & Arn. [48], 19.45% for the leaf methanolic extract of *Momordica angustisepala* Harms and 15.32% for the leaf methanolic extract of *Drynaria laurentii* (Christ) Hieron. [49].

Better results were obtained by Marrelli et al. (2022) for the n-hexane fractionated EE of *Allium cepa* L. var. Tropea, obtaining a relative abundance of 29.4% [50]. For *Z. irwiniana* even though root extracts have shown better results, for commercial production the best option is to use leaves collected from individuals grown in a greenhouse, as they presented a relative abundance of 11.62%, close to the maximum obtained, with the advantage of using the individual in a sustainable way by collecting only the aerial parts.

Lycorine is an alkaloid with immense therapeutic potential [51], widely found in the bulbs and leaves of plants from the Amaryllidaceae family [52]. The results of previous studies showed that among the groups of alkaloids extracted from this family, representatives of the galantamine and lycorine groups presented a significantly higher AChE inhibitory potential than the others [53,54,55]. Furthermore, other properties such as antitumor, bactericidal, antiparasitic [51], and anticancer [56] activities have been described. Previous studies have also produced a series of derived structures to explore the relationship between their structure and biological activity [19,52]. These facts strongly motivate the screening of Amaryllidaceae alkaloids, comprising different structural types for their AChE inhibitory activity and other biological activities.

Some studies classify lycorine as one of the main alkaloids in terms of relative abundance in the leaves of Amaryllidacae species [9,31,57,58] and in the bulbs. Among these, Andrade et al. (2012b) found 2.35% for *Narcissus broussonetii* Lag. [59] and in a later study, they found 9.26% for *Hippeastrum aulicum* (Ker Gawl.) Herb. and 41.89% for *Hippeastrum calyptratum* (Ker Gawl.) Herb. [60], all in ethyl acetate fractions from bulbs. The best relative abundance of lycorine obtained in the present study for *Z. irwiniana* was in the leaves of individuals propagated in a culture medium with a concentration of 60 gL^−1^ of sucrose, without the use of AC (2,57%). Paiva et al. (2020) found the alkaloids galantamine and pseudolycorine in extracts obtained from *Z. irwiniana* bulbs at concentrations of 41.7 ± 0.9 and 372.6 ± 28.2 µg g^−1^ [8]. In agreement with Tasker et al. (2018), lycorine was not detected in the root [52].

Previous studies have identified lycorine and other alkaloids in the bulbs and aerial parts of *Zephyranthes concolor* (Lindl.) Benth & Hook. f. [61], *Zephyranthes grandiflora* Lindl. [62], *Zephyranthes robusta* (Herb.) Baker [63], *Zephyranthes candida* (Lindl.) Herb. [64,65,66,67], and *Zephyranthes citrina* Baker [68]; however, these studies aimed for, in common with each other, the identification and structural elucidation of the alkaloids of these species and not the relative abundance in the bulbs and leaves.

Ortiz et al. (2012) found different abundances for lycorine in the basic chloroform fraction of bulbs of *Zephyranthes jamesonii* (Baker) Nic. García & S.C. Arroyo when comparing different collection locations [10]. Centeno-Betanzos et al. (2022) found a similar result in bulb ethyl acetate fractions of *Zephyranthes alba* Flagg, G. Lom. Sm. & García-Mend. and *Zephyranthes fosteri* Traub, where the relative abundance of lycorine also varied depending on the collection site, respectively, from 4.28 to 8.45% and 46.36 to 88.82% [69].

These results and those obtained in the present study (leaf: T1 0.87%; T5 2.57%; T6 0.86%; GH 0.54%; bulb: T1 0.74%; T3 0.32%; T6 1.09%; GH 0.59%) indicate that lycorine production is variable and depends on the substrate or culture medium, and that *in vitro* propagation is a viable alternative to produce lycorine, because individuals propagated in a greenhouse presented the lowest relative abundances in relation to the other treatments.

Due to the small amount of obtained plant material from micropropagation, the yield of the extracts was not sufficient enough to undertake additional experiments to achieve a better chemical profile of them. Moreover, this study revealed that an extended experiment could lead to better information regarding the supplementation. The histochemical location of the alkaloids, in the leaves and bulb of *Z. irwiniana*, previously unknown, was presented in this study. Determining the localization of secondary compounds in plant tissues can provide a better understanding of the role and sites of synthesis and accumulation of these compounds [15]. However, data on the histolocalization of alkaloids in Amaryllidaceae are scarce and there is a gap in our knowledge about the secondary metabolites of this family. Histochemical tests with Dragendorff’s and Wagner’s reagents identified alkaloids in the chlorenchyma cells of the leaf margins and in the bulbs. Similar results were obtained for some species of the Fabaceae family, proposing that green tissues, particularly the palisade mesophyll of the leaves, are the main source of alkaloids [15]. Silva et al. (2013) observed that almost all mesophyll cells in the leaves of *Crinum americanum* L. presented alkaloid content [70], different from the species *Z. irwiniana*, where only the chlorenchymatic cells presented alkaloids.

## 4. Materials and Methods

### 4.1. Plant Material

*Zephyranthes irwiniana* plants (bulbs) were obtained from the Amaryllidaceae Germplasm Collection at Embrapa Genetic Resources and Biotechnology—CENARGEN. Bulbs were cultivated and voucher deposited in Herbarium UB (Fagg CW 2556). Seeds collected and stored in brown paper bags for 3 months at room temperature until sowing. Non-viable seeds were eliminated, identified by their small size and reduced reserve tissue.

### 4.2. Germination and Cultivation in a Greenhouse

Germination of *Z. irwiniana* seeds was carried out in June 2021, using 162 tubes measuring 280 cm^3^ (190 mm × 63 mm) with Bioplant Plus^®^ commercial substrate pH 6.2, containing one seed per tube. After sowing, the tubes were placed in the greenhouse of the Reference Center for Nature Conservation and Recovery of Degraded Areas, located at the University of Brasília, Campus Darcy Ribeiro (Lat: 15°46′16.56″ S and Long: 47°52′3.78″ O), and covered with 50% shading Sombrite^®^, under natural temperature and light conditions. The beginning of germination was observed through the emission of the first leaf. Watering was carried out once a day using an automatic system, programmed to irrigate every day for 40 min at 7:00 am in the morning. Air temperature and humidity were recorded daily with a data logger, with a measurement frequency of every minute, for the monthly calculation of the average, maximum and minimum of both. The seedlings resulting from germination were cultivated for a period of 270 days until collection in March 2022, and they were identified as GH treatment. It is important to note that the cultivation in the greenhouse was carried out solely to observe the seeds behavior when subjected to two different circumstances.

### 4.3. In Vitro Germination

To prepare the culture media, MS culture medium (Sigma-Aldrich M5519, St. Louis, MO, USA) was used, supplemented with 3% sucrose (Macron Fine Chemicals^TM^) [71], and 0.15% Phytagel^®^. The pH of the culture medium was adjusted to 5.8. Then, it was distributed in the amount of 20 mL in 25 × 150 mm test tubes. Subsequently, the test tubes containing the culture medium were sterilized in an autoclave at a temperature of 121 °C, under a pressure of 1.5 atm, for a period of 20 min. A total of 154 seeds were disinfected with an ethanol solution (70%) for 1 min, followed by immersion in a commercial solution of sodium hypochlorite (2–2.5%) for 8 min and, subsequently, washed three times with sterile distilled water. After this, the seeds were inoculated and kept in a growth room with a lighting intensity of 50 µmol.m^−2^s^−1^, for a photoperiod of 12 h, for 120 days. The beginning of germination was observed from root emergence.

### 4.4. Micropropagation of Bulblets

To prepare the culture media for the 6 treatments, MS culture medium [71] (Sigma-Aldrich M5519, St. Louis, MO, USA) was used, supplemented with 0.15% Phytagel^®^. Furthermore, the culture media from treatments T1 to 2, T3 to 4 and T5 to 6 were supplemented with the respective sucrose concentrations of 30 gL^−1^, 45 gL^−1^ and 60 gL^−1^ (Macron Fine Chemicals^TM^, Radnor Township, PA, USA), and the T2, T4 and T6 treatment media were also supplemented with 1.0 gL^−1^ of activated charcoal (Sigma-Aldrich C9157, USA). Culture medium pH was adjusted to 5.8, and 20 mL was added to the 25 × 150 mm test tubes which were then sterilized in an autoclave at 121 °C, under a pressure of 1.5 atm, for 20 min.

After 120 days of *in vitro* germination of *Z. irwiniana* seeds, the test tubes were randomly separated into 6 groups, each with 25 seedlings corresponding to the same treatment. Each seedling had its root and leaves separated from the bulblet and the bulblet was inserted into a new test tube containing the corresponding treatment (T1–T6), where the treatment T1 is the control. Finally, the test tubes were closed and placed in the growth room with a lighting intensity of 50 µmol m^−2^ s^−1^, for a photoperiod of 12 h.

The bulblets were cultivated for a period of 150 days, and at the end of this period (March 2022), all seedlings and plants cultivated, *in vitro* and in a greenhouse, were collected and washed with running water.

### 4.5. Seedling Growth Analysis

The length of the main root and leaf, the number of roots and leaves, and the total fresh and dry biomass of each seedling were evaluated. The number of roots and leaves was determined by simple counting. In relation to the individuals that had bulblets in addition to the main bulb, these were measured individually (length and quantity of root and leaf); however, to obtain the fresh and dry biomass, they were weighed together, as they originated from the same seed. To obtain fresh and dry biomass, each seedling was weighed using an analytical balance.

### 4.6. Zephyranthes irwiniana Extract

The leaves, bulbs and roots of *Z. irwiniana* were dried in an oven with air circulation at 40 °C for 120 h. Plant material was homogenized and macerated in hexane P.A for 72 h. After the third maceration with hexane, a second extraction was carried out with ethanol for 72 h, and after the third maceration with ethanol, the extracted solution was filtered and concentrated at 40 °C using a rotary evaporator under vacuum (Hei-VAP Advantage, ML, G1, 115v—Heidolph, Schwabach, Germany). Ethanolic extract (EE) was stored in a freezer at −20 °C to be further analyzed through GC–MS. Due to the small amount of material obtained from the extraction with ethanol (crude extract), it was not possible to subsequently carry out the acid-base extraction to obtain the hexane, ethyl acetate and ethyl acetate–methanol fractions.

### 4.7. GC–MS Analysis

A total of 8 mg of EE was dissolved in 1 mL of MeOH and injected directly into the GC–MS apparatus (Clarus 680 GC, Perkin Elmer) coupled to a quadrupole mass spectrometer (Clarus SQ8 MS, Perkin Elmer, Singapore). A Perkin Elmer Elite-5MS capillary column (length 30 m × inner diameter 0.25 mm × film thickness 0.25 µm) was used. The temperature gradient was performed as follows: 12 min at 100 °C, 100–180 °C at 15 °C/min, 180 at 300 °C at 5 °C/min, and 10 min at 300 °C. The injector and detector temperatures were 280 and 250 °C, respectively, and the carrier gas flow rate (He) was 1 mL/min. The injection volume was 1 µL. Alkaloids were identified by comparing their mass spectra and retention index (RI). Mass spectra were analyzed using AMDIS 2.64 software (NIST) (Gaithersburg, MD, USA), and RI was recorded with a calibration mixture of hydrocarbon standards (C9–C36). The proportion of each alkaloid present in extracts and fractions analyzed through GC–MS was expressed as a percentage of the alkaloid peak area as a function of the total ion current (TIC).

### 4.8. Compounds Identification

The identification was accomplished by comparing the Kovats retention index and the mass spectrometric data (molecular ion peaks and fragmentation patterns), to those recorded in the MS library software (NIST Mass Spectral Search Program for the NIST/EPA/NIH Mass Spectral library, 2014 version 2.2).

### 4.9. Histolocalization of Alkaloids in the Leaf and Bulb of Zephyranthes irwiniana

Histochemical tests with Dragendorff’s [72] and Wagner’s [73] reagents were carried out on fresh sections of leaves and bulbs, and cut by hand, from 4 individuals from the T1 treatment (30 gL^−1^ sucrose without activated charcoal). The Dragendorff’s reagent stock solution was prepared with 25 mL of 12.5% bismuth nitrate in 25% acetic acid, and 10.0 mL of 40% potassium iodide. For use, 5 mL of the solution was removed and supplemented with 10 mL of acetic acid, making up to 100 mL with distilled water. After preparing the reagent for use, the anatomical sections were in contact with the reagent for a period of 5 to 10 min, then quickly washed with 5% sodium nitrite and then washed in distilled water. The preparation of Wagner’s reagent was carried out using 2 g of potassium iodide and 1.27 g of iodine, dissolved in 100 mL of distilled water. The anatomical sections were in contact with the reagent for a period of 5 to 10 min and then washed with distilled water. The presence of alkaloid was identified by the reddish-brown color for both reagents. For the control, fresh sections of the leaves and bulbs, cut by hand, from the same 4 individuals from the T1 treatment (control) were prepared without contact with the Dragendorff and Wagner reagent.

### 4.10. Statistical and Data Analysis

The seedling growth data were tested via two-way analysis of variance (ANOVA) followed by Tukey’s test (Sisvar Software, version 5.6), and *p* < 0.05 was considered significant. For single samples, the t test (Graph Pad Prism Software, version 9.0) was used and *p* < 0.05 was considered significant.

## 5. Conclusions

For *Zephyranthes irwinana* the constitution of the culture medium directly influences plant growth and phytochemistry. Its great contribution is that variations in sucrose concentration and the use of AC can be tested for other species of commercial interest, since the results obtained for *Z. irwiniana* are promising, especially regarding the increase in biomass production and abundance of secondary metabolites, depending on the adjustment of the culture medium.

Studies are necessary to elucidate the mechanisms of interaction between sucrose and AC in plant growth and phytochemistry, as it is not clear how these interactions work, especially in relation to phytochemistry, as these results did not present a detectable pattern.

Finally, the results demonstrate that adjustments in the culture medium are effective in obtaining better results in plant growth and phytochemistry, according to the established objective. However, this is a complex process and requires further research to define the relationships between the biosynthesis of bioactive compounds and the optimization of the culture medium [74].

## Figures and Tables

**Figure 1 plants-13-00569-f001:**
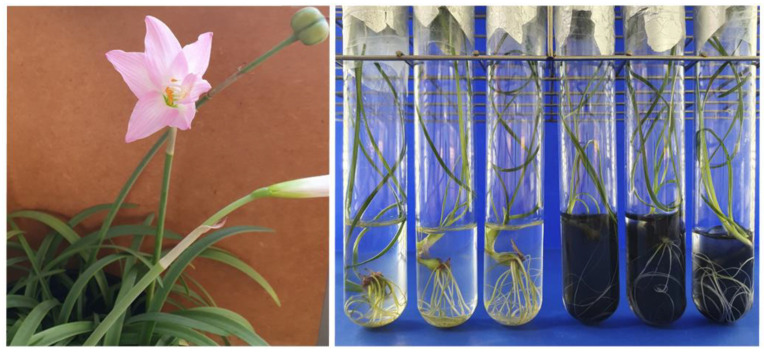
*Zephyranthes irwiniana* (Ravenna) Nic. García. (**Left**): flower and fruit details. (**Right**): micropropagation with sucrose and activated charcoal.

**Figure 2 plants-13-00569-f002:**
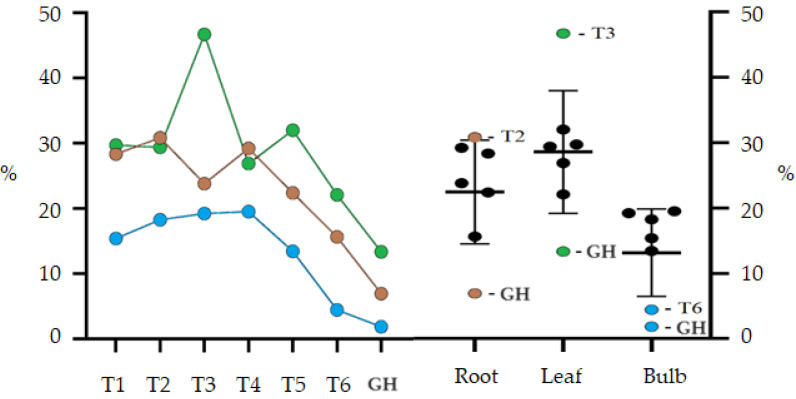
Percentage yield of ethanolic extract (EE) from root, leaf and bulb of *Zephyranthes irwiniana* cultivated *in vitro* and greenhouse via treatment. Blue line: Bulb. Green line: Leaf. Brown line: Root. The bars are the confidence interval (95%).

**Figure 3 plants-13-00569-f003:**
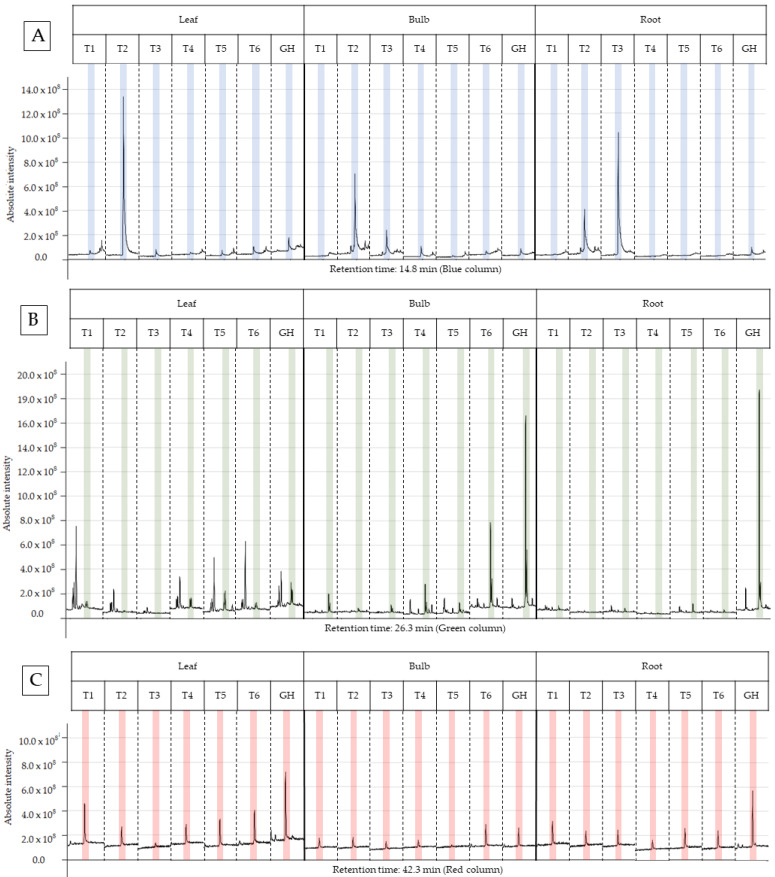
Peak of the compound in the chromatogram of each ethanolic extract from treatments T1 to T6 for leaf, bulb and root of *Zephyranthes irwiniana* using micropropagation and GH for leaf, bulb and root of *Z. irwiniana* cultivated in greenhouse. (**A**) α-terpinyl-acetate (blue column). (**B**) ethyl linoleate (green column). (**C**) clionasterol (red column).

**Figure 4 plants-13-00569-f004:**
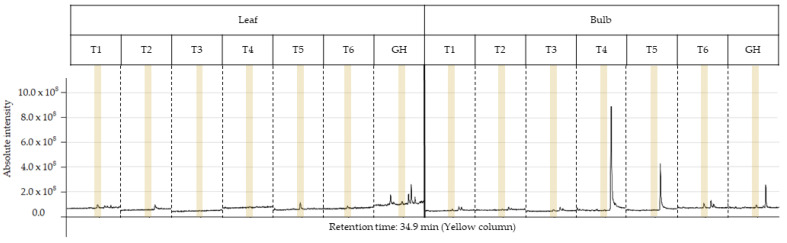
Peak of the lycorine (yellow column) in the chromatogram of each ethanolic extract from treatments T1 to T6 for leaf and bulb of *Zephyranthes irwiniana* using micropropagation and GH for leaf and bulb of *Z. irwiniana* cultivated in greenhouse.

**Figure 5 plants-13-00569-f005:**
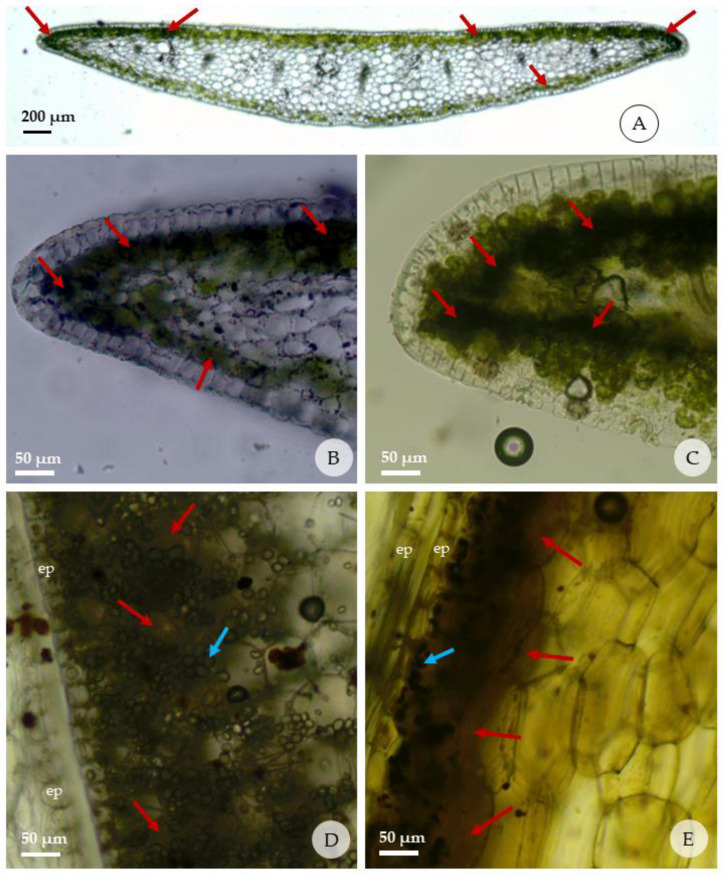
Reactions of fresh leaves and bulbs of *Zephyranthes irwiniana* (cross sections) submitted to the histochemical tests; red arrows indicate presence of alkaloids; blue arrows indicate the starch grains. (**A**–**C**) detail of presence of alkaloids in the leaf margins, just below the adaxial and abaxial epidermis; (**D**,**E**) detail of presence of alkaloids in the bulbs in the region adjacent to the epidermis that faces the outside of each cataphyll; (**A**,**B**,**D**) Dragendorff’s test; (**C**,**E**) test with Wagner’s reagent. (ep: epidermis).

**Table 1 plants-13-00569-t001:** Results of vegetative growth data collected from *Zephyranthes irwiniana*.

	(gL^−1^)	Treatment	Leaf nº	Root nº	Leaf le. (cm)	Root le. (cm)	Fresh biom. (g)	Dry biom. (g)
Sucrose	30.0	T1-T2	3.78 a	5.33 a	20.25 a	16.83 a	0.44 a	0.05 a
	45.0	T3-T4	3.86 a	7.97 b	20.33 a	18.19 a	0.66 b	0.06 a
	60.0	T5-T6	4.02 a	10.89 c	19.53 a	17.64 a	0.83 c	0.10 b
AC	0.0	T1, T3, T5	3.63 a	8.89 a	18.31 a	14.95 a	0.61 a	0.06 a
	1.0	T2, T4, T6	4.15 b	7.24 b	21.76 b	20.16 b	0.68 a	0.07 a

Leaf nº: leaf number. Root nº: root number. Leaf le.: leaf length. Root le.: root length. Fresh biom.: fresh biomass. Dry biom.: dry biomass. AC: activated charcoal. T1 is control. Different letters following the mean values in columns indicate significant differences via Tukey’s test (*p* < 0.05).

**Table 2 plants-13-00569-t002:** Results of sucrose breakdown within each AC level in vegetative growth data collected from *Zephyranthes irwiniana*.

AC (gL^−1^)	Sucrose (gL^−1^)	Treatment	Leaf nº	Root nº	Leaf le. (cm)	Root le. (cm)	Fresh biom. (g)	Dry biom. (g)
0.0	30.0	T1	3.67 a	6.11 a	18.89 a	13.71 a	0.47 a	0.04 a
	45.0	T3	3.61 a	8.33 b	18.91 a	16.55 b	0.64 ab	0.06 a
	60.0	T5	3.66 a	12.22 c	17.13 b	14.59 ab	0.71 b	0.09 b
1.0	30.0	T2	3.89 a	4.56 a	21.60 a	19.96 a	0.40 a	0.05 a
	45.0	T4	4.11 a	7.61 b	21.74 a	19.83 a	0.67 b	0.06 a
	60.0	T6	4.44 a	9.56 b	21.93 a	20.69 a	0.96 c	0.11 b

Leaf nº: leaf number. Root nº: root number. Leaf le.: leaf length. Root le.: root length. Fresh biom.: fresh biomass. Dry biom.: dry biomass. AC: activated charcoal. T1 is control. Different letters following the mean values in columns indicate significant differences via Tukey’s test (*p* < 0.05).

**Table 3 plants-13-00569-t003:** Results of AC breakdown within each sucrose level in vegetative growth data collected from *Zephyranthes irwiniana*.

AC (gL^−1^)	Sucrose (gL^−1^)	Treatment	Leaf nº	Root nº	Leaf le. (cm)	Root le. (cm)	Fresh biom. (g)	Dry biom. (g)
0.0	30.0	T1	3.67 a	6.11 a	18.90 a	13.71 a	0.47 a	0.04 a
1.0		T2	3.89 a	4.56 a	21.60 b	19.96 b	0.40 a	0.05 a
0.0	45.0	T3	3.61 a	8.33 a	18.91 a	16.55 a	0.64 a	0.06 a
1.0		T4	4.11 b	7.61 a	21.75 b	19.83 b	0.67 a	0.06 a
0.0	60.0	T5	3.61 a	12.22 a	17.13 a	14.59 a	0.71 a	0.09 a
1.0		T6	4.44 b	9.56 b	21.93 b	20.69 b	0.96 b	0.11 a

Leaf nº: leaf number. Root nº: root number. Leaf le.: leaf length. Root le.: root length. Fresh biom.: fresh biomass. Dry biom.: dry biomass. AC: activated charcoal. T1 is control. Different letters following the mean values in columns indicate significant differences via Tukey’s test (*p* < 0.05).

**Table 4 plants-13-00569-t004:** Compounds of *Zephyranthes irwiniana* identified using gas chromatography–mass spectrometry (GC–MS).

Treat.	%TIC	RT (min)	RI (exp)	RI (Lit)	Compound	Mass Fragmentation (Relative Intensity)
T3 (root)	40.8	14.9	1351	1350 ± 3	α-Terpenyl acetate	121 (100), 93 (62), 43 (47), 136 (39), 67 (28), 68 (22), 107 (20), 91 (17), 79 (16), 77 (15)
T1 (leaf)	2.9	15.7	1390	1396 ± 2	Ethyl decanoate	88 (100), 101 (51), 70 (29), 73 (39), 157 (25), 41 (25), 61 (24), 155 (22), 60 (21), 43 (20)
T2 (leaf)	1.7	17.3	1499	1499 ± 8	Eremophylene	55 (100), 107 (97), 189 (96), 81 (88), 79 (82), 105 (73), 108 (57), 121 (55), 133 (54), 161 (41)
T1 (bulb)	3.5	18.3	1590	1595 ± 2	Ethyl decanoate	88 (100), 101 (38), 73 (26), 70 (23), 41 (21), 43 (20), 55 (18), 69 (14), 57 (13), 160 (12)
T1 (leaf)	4.7	21.4	1832	1837 ± 5	Neophytadiene	68 (100), 69 (70), 82 (63), 95 (56), 57 (53), 67 (49), 71 (44), 81 (40), 55 (38), 41 (36)
T6 (bulb)	3.7	22.7	1922	1926 ± 2	Methyl palmitate	74 (100), 87 (47), 43 (15), 75 (14), 143 (11), 227 (9), 83 (6), 59 (5), 129 (5), 171 (5)
T2 (bulb)	1.4	23.2	1956	1968 ± 7	Palmitic acid	60 (100), 43 (92), 41 (81), 55 (65), 129 (64), 69 (61), 83 (47), 213 (33), 185 (28), 256 (27)
T6 (bulb)	12.3	23.7	1990	1993 ± 3	Ethyl palmitate	88 (100), 70 (27), 43 (23), 41 (19), 55 (18), 57 (16), 69 (15), 89 (15), 157 (15), 241 (11)
GH (root)	4.0	25.2	2090	2092 ± 4	Methyl linoleate	81 (100), 67 (70), 68 (47), 41 (41), 82 (33), 96 (31), 69 (28), 79 (27), 64 (24), 109 (21)
T1 (leaf)	6.4	25.3	2095	2098 ± 3	Methyl linolenate	79 (100), 67 (83), 95 (56), 108 (48), 80 (48), 93 (47), 55 (43), 69 (28), 77 (28), 107 (26)
GH (root)	38.4	26.3	2156	2162 ± 6	Ethyl linoleate	67 (100), 81 (94), 95 (57), 82 (48), 55 (48), 79 (43), 41 (42), 68 (42), 69 (35), 109 (32)
T5 (leaf)	7.0	26.4	2163	2171 ± 13	Ethyl linolenate	79 (100), 95 (82), 81 (62), 108 (40), 41 (40), 44 (32), 55 (29), 93 (25), 149 (17), 119 (16)
T5 (bulb)	28.6	29.3	2353	2374 ± 25	Octadecanamide	59 (100), 72 (79), 41 (31), 55 (30), 69 (24), 43 (22), 83 (15), 67 (14), 81 (13), 126 (12)
T5 (leaf)	2.6	34.9	2734	2747 − N/A	Lycorine	226 (100), 250 (62), 227 (52), 287 (23), 268 (19), 286 (16), 228 (10), 240 (6), 269 (6), 270 (5)
GH (leaf)	11.6	42.3	3310	3321 ± 31	Clionasterol	43 (100), 107 (88), 145 (82), 414 (77), 55 (77), 81 (75), 161 (72), 95 (72), 213 (68), 93 (66)

Identification was based on comparison of the compounds mass spectral data (MS) and retention indices (RI) with those of the NIST Mass Spectral Library (2014). The proportion of each compound is expressed as a percentage (%) of the total compounds measured by total ion current (TIC). The TIC% corresponds to the selected treatment (previous column) that obtained the best peak resolution. Treat.: treatment. RT: retention time. RI (exp): retention index calculated. RI (Lit): published retention index (NIST, 2014).

**Table 5 plants-13-00569-t005:** Intensity of the reaction of the fresh leaves and bulbs of *Zephyranthes irwiniana* (cross sections) submitted to the histochemical tests.

Compound	Tests	Leaves	Bulbs
Alkaloid	Dragendorff	+	+
	Wagner	++	++

Note: (+) moderate reaction; (++) strong reaction.

## Data Availability

The original contributions presented in the study are included in the article, further inquiries can be directed to the corresponding author.

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

*in vitro* systems. Process Biochem..

