# Peer review of "Influence of Sucrose and Activated Charcoal on Phytochemistry and Vegetative Growth in Zephyranthes irwiniana (Ravenna) Nic. García (Amaryllidaceae)"

_plants, 2024, doi:10.3390/plants13050569_

Round 1

Reviewer 1 Report

Comments and Suggestions for Authors

This paper reports the Influence of Sucrose and Activated Charcoal on Phytochemistry and Vegetative Growth in Zephyranthes irwiniana (Ravenna) Nic. García (Amaryllidaceae). The manuscript is well prepared and the subject is interesting. In my opinion, the manuscript is in a position to be accepted for publication after minor revision. Here are some comments on the manuscript:

-       The authors must revise the cited keywords. Some words like alkaloid’s histolocalization; Cerrado; acetylcholinesterase are not cited in the abstract.

-       In the introduction part, the authors should add some information’s concerning the biological properties, traditional and medicinal uses of the studied specie in Brazil.

-       The authors must add a conclusion section to this paper.

-       There are some missing and extra space issues. Kindly go through the manuscript again.

-       Check the whole paper to remove spelling mistakes and insert blanks where necessary.

Author Response

We are very grateful for your pertinent comments. The answers for each comment is given below:

  1. The authors must revise the cited keywords. Some words like alkaloid’s histolocalization; Cerrado; acetylcholinesteraseare not cited in the abstract.

We appreciate your contributions. We replaced these keywords with histolocalization test; vegetative growth and phytochemistry and Cerrado was added to the abstract. 

  1. In the introduction part, the authors should add some information’s concerning the biological properties, traditional and medicinal uses of the studied specie in Brazil

We added this information in lines 43 - 45. “Few studies have evaluated the biological properties of this species [8], but there is no record of its traditional and medicinal use in Brazil.” 

  1. The authors must add a conclusion section to this paper.

The text was modified and added to the conclusion section (See lines 597 – 619).

  1. There are some missing and extra space issues. Kindly go through the manuscript again.

Thank you for your suggestions, all were undertaken. 

  1. Check the whole paper to remove spelling mistakes and insert blanks where necessary.

The text was adjusted in accordance with the suggestion. Thank you for your suggestions, all were undertaken.

Reviewer 2 Report

Comments and Suggestions for Authors

The article entitled “Influence of Sucrose and Activated Charcoal on Phytochemistry and Vegetative Growth in Zephyranthes irwiniana (Ravenna) Nic. García (Amaryllidaceae)” fits with the general scope of the journal Plants. In this manuscript, the authors aimed at investigating the vegetative growth and phytochemistry of some bioactive compounds in Zephyranthes irwiniana, in response to variations in sucrose concentration and activated carbon added in the culture medium. The manuscript is well written, the experiments were well-designed, and there were multiple results. The work could be useful for growers; however, there are several minor issues that need to be corrected before publication.

Rows 82 – Tables 1–3

Data presented in Tables 1 - 3 should be indicated as mean ± SD (standard deviation).

Please, be consistent when adding letters after the value in tables.

Activated carbon has a great deal in common with charcoal but there are some differences. While charcoal is traditionally made from wood, activated carbon may be made from wood, peat, nutshells, coconut husks, lignite, coal, coir, or petroleum pitch. It is well-known that activated charcoal is often used in tissue culture to improve cell growth and development. It plays a critical role in micropropagation, seed germination, etc. One should give more information about the activated carbon used in experiments.

One should write the format of the references according to the journal’s recommendations. Some references use capital letter of each word (in the titles of the article), whereas others use small letters. One should check the denomination of the journal (abbreviated or complete name) - some references preserve the abbreviations of the denomination of the journals whereas others keep their complete denominations.

Denomination abbreviated of journals, preferred by MDPI journals, are: Sci. Hortic., Plant Cell Tiss. Organ Cult., Neurochem. Int., In Vitro Cell.Dev.Biol.-Plant, Biotechnol. Biotechnol. Equip., Nat. Prod. Commun., Plant Growth Regul., etc.

Because the cited articles are numbered, there is no need to insert the letter a and b, e.g. 2013a, 2013 b, for the same principal author (let's look at references 15 and 16 or 18 and 19).

Authors must be cited correctly: 2. Rahimi Khonakdari, M.; 8. da Costa, G.G.P.; 16. Herranz, J. M.; Copete, M. A.; Ferrandis, P.;  Herranz, J.M.; Copete, M.A.; Ferrandis, P.; 22. El Tahchy, A.

Author Response

We are very grateful for your pertinent comments. The answers for each comment is given below:

  1. Rows 82 – Tables 1–3. Data presented in Tables 1 - 3 should be indicated as mean ± SD (standard deviation). Please, be consistent when adding letters after the value in tables.

Thank you for your comments. Yes, they are mean values and we have added that to the table legends. The standard deviation values show the differences between mean values but do not give the probability that the values are significant at 5% which the Tukey test does. The mean comparison test was used in our experiment because it presents an important factor, which is probability. Therefore, the results show a level of security to the point of affirming that, indeed, the averages differ from each other statistically. An example of this is that, when a Tukey test is carried out, after analyzing the data, we will be given a 5% probability of the results, therefore, it gives us a level of security to make a decision. For this reason, the research team decided to make use of this statistical analysis by comparing means, choosing the Tukey test because it is suitable for the design of the experiment and that, in addition to being versatile, it can be used to test any and all contrasts between two treatment averages. Letters after the means are used to differentiate, and different letters refer to means that are statistically different from each other in the columns, and the same letters that there is no difference between the means (treatments).

  1. Activated carbon has a great deal in common with charcoal but there are some differences. While charcoal is traditionally made from wood, activated carbon may be made from wood, peat, nutshells, coconut husks, lignite, coal, coir, or petroleum pitch. It is well-known that activated charcoal is often used in tissue culture to improve cell growth and development. It plays a critical role in micropropagation, seed germination, etc. One should give more information about the activated carbon used in experiments.

We appreciate your comments. We added the reference for activated charcoal (Sigma- Aldrich, C9157) in the line 548. We assessed the safety data sheet and the product information (https://www.sigmaaldrich.com/BR/en/product/sial/c9157) and there is no information regarding the charcoal origin.

  1. One should write the format of the references according to the journal’s recommendations. Some references use capital letter of each word (in the titles of the article), whereas others use small letters. One should check the denomination of the journal (abbreviated or complete name) - some references preserve the abbreviations of the denomination of the journals whereas others keep their complete denominations.

Denomination abbreviated of journals, preferred by MDPI journals, are: Sci. Hortic., Plant Cell Tiss. Organ Cult., Neurochem. Int., In Vitro Cell.Dev.Biol.-Plant, Biotechnol. Biotechnol. Equip., Nat. Prod. Commun., Plant Growth Regul., etc.

Because the cited articles are numbered, there is no need to insert the letter a and b, e.g. 2013a, 2013 b, for the same principal author (let's look at references 15 and 16 or 18 and 19).

Authors must be cited correctly: 2. Rahimi Khonakdari, M.; 8. da Costa, G.G.P.; 16. Herranz, J. M.; Copete, M. A.; Ferrandis, P.;  Herranz, J.M.; Copete, M.A.; Ferrandis, P.; 22. El Tahchy, A. 

Many thanks, the references were adjusted in accordance with the suggestions (See lines 643 -809)

Reviewer 3 Report

Comments and Suggestions for Authors

Well presented research, but there are some things that need clarifying / correcting or that can be improved.

Introduction:

Explain in the introduction why you are testing the effect of sucrose and activated charcoal added to the growth medium on growth / phytochemistry. What do you expect this will change, and why? Is there literature you can reference for this?

Is an objective also to compare greenhouse versus in vitro germination?

Explain what AD means when you first use it. It's explained in the abstract, but not in the text.

Results:

Is there an explanation for the very low germination rate in greenhouse? Is this typical for the species / family?

What is the meaning of a en b in Tables 1-3? It's about significant differences but what does a mean and what b. In these tables you present mean values but don't mention it. These are presumably the means of your repetitions. And why no standard deviation values? 

Can Table 2 and 3 not be merged? As far as I see they contain the same data, just organised differently.

In 2.3 you introduce treatment T7, but nowhere in the paper you explain which treatment this is. Is this the control? Are these the plants grown in greenhouse? Tables 1-3 contain only T1 to T6; Why not T7?

In 2.4 you state that the selected compounds have characteristics of interest according to literature, yet you give no literature reference. Or do you mean the literature results in the discussion? if so, then clarify that here. 

Discussion:

In paragraph 2 you state that greenhouse cultivation is more advantageous for increasing biomass, especially for the bulb. I have to search well to find this information in the results. Make it more explicit, eg by repeating the numbers here, or by pointing it out in the results.

Also indicate limitations of your study in the discussion. Every study has limitations;

Materials & Methods:

Why is there no control used in the study, where you do not add extra sucrose / charcoal? Or is this the T7 that is not explained? Or is the greenhouse experiment the control?

Explain T7.

Supplementary materials:

Explain in this file what the figures mean.

Author Response

Many thanks for you pertinent comments which are replied below:

  1. Introduction: Explain in the introduction why you are testing the effect of sucrose and activated charcoal added to the growth medium on growth / phytochemistry. What do you expect this will change, and why? Is there literature you can reference for this?

We added this information in lines 60 - 61. “We expect that in addition of sucrose effects, the activated charcoal can modify the biomass increase rate and the phytochemistry due to its adsorption capacity [14].”

Thomas, T.D. The role of activated charcoal in plant tissue culture. Biotechnol. Adv. 2008, 26, 618-631. [CrossRef]

We added this information in lines 56- 58. “According to the literature [13], increasing the concentration of sucrose in the culture medium up to 90 gL-1 increases the biomass of some Amaryllidaceae species.”

Ptak, A.; MoraÅ„ska, E.; Skrzypek, E.; Warchol, M.; Spina, R.; Laurian-Mattar, D.; Simlat, M. Carbohydrates stimulated Amaryllidaceae alkaloids biosynthesis in Leucojum aestivum L. plants cultured in RITA® bioreactor. PeerJ 2020, 8, e8688. [CrossRef] 

  1. Introduction: Is an objective also to compare greenhouse versus in vitro germination?

No, germination in a greenhouse is just a parallel experiment to observe with the development of plants from the micropropagation. The text was altered for greater clarity.

  1. Introduction: Explain what AD means when you first use it. It's explained in the abstract, but not in the text.

We added this information in line 40.

  1. Results: Is there an explanation for the very low germination rate in greenhouse? Is this typical for the species / family?

Germination in the greenhouse had different substrate and edaphic conditions from the micropropagation experiment, including reaching higher temperatures which in a related species from the Brazilian caatinga (Zephyranthes sylvatica) showed thermo-dormancy in seeds above 25C (Wesley de Silva et al. 2014). This is one reason for further using micropropagation for Z. irwiniana as an alternative for direct seeding in horticulture.

Wesley da Silva, Laise Guerra Barbosa, José Eduardo Santos Barbosa da Silva, Keylan Silva Guirra, Diego Rangel da Silva Gama, Gilmara Moreira de Oliveira, Bárbara França Dantas. Caracterization of seed germination of Zephranthes sylvatica (Mart.) Baker (Amarilidacea) Journal of Seed Science 2014, 36, 2, 178-185.

  1. Results: What is the meaning of a en b in Tables 1-3? It's about significant differences but what does a mean and what b. In these tables you present mean values but don't mention it. These are presumably the means of your repetitions. And why no standard deviation values?

Thank you for your comments. Yes, they are mean values and we have added that to the table legends (see line 102, 108 and 114). The standard deviation values show the differences between mean values but do not give the probability that the values are significant at 5% which the Tukey test does. The mean comparison test was used in our experiment because it presents an important factor, which is probability. Therefore, the results show a level of security to the point of affirming that, indeed, the averages differ from each other statistically. An example of this is that, when a Tukey test is carried out, after analyzing the data, we will be given a 5% probability of the results, therefore, it gives us a level of security to make a decision. For this reason, the research team decided to make use of this statistical analysis by comparing means, choosing the Tukey test because it is suitable for the design of the experiment and that, in addition to being versatile, it can be used to test any and all contrasts between two treatment averages. Letters after the means are used to differentiate, and different letters refer to means that are statistically different from each other in the columns, and the same letters that there is no difference between the means (treatments).

  1. Results: Can Table 2 and 3 not be merged? As far as I see they contain the same data, just organized differently.

The data in tables 2 and 3 are similar, however table 2 evaluates whether there was an influence of the sucrose breakdown within each AC level (0.0 or 1.0 gL-1) and table 3 evaluates whether there was an influence of activated charcoal breakdown within each sucrose level (30, 45 and 60 gL-1) using the Tukey test. The data are similar, but the tables compare the means differently to indicate whether there was a difference between the means.

  1. Results: In 2.3 you introduce treatment T7, but nowhere in the paper you explain which treatment this is. Is this the control? Are these the plants grown in greenhouse? Tables 1-3 contain only T1 to T6; Why not T7?

Thank you very much for highlighting this issue. The T7 treatment refers to the plants that grew in the greenhouse, therefore we have altered the text to replace T7 for GH for better clarity considering germination in a greenhouse was just a parallel experiment to compare with the development of plants from the micropropagation. That’s why T7 (now GH) was not included in Tables 1-3. The control is referred in T1, with 3% of glucose (30 g/L).

  1. Results: In 2.4 you state that the selected compounds have characteristics of interest according to literature, yet you give no literature reference. Or do you mean the literature results in the discussion? if so, then clarify that here.

We added the literature reference in line 169.

  1. Discussion: In paragraph 2 you state that greenhouse cultivation is more advantageous for increasing biomass, especially for the bulb. I have to search well to find this information in the results. Make it more explicit, eg by repeating the numbers here, or by pointing it out in the results.

Many thanks for the suggestion. We added this information in green highlight to see it better.

  1. Discussion: Also indicate limitations of your study in the discussion. Every study has limitations.

We added this information in lines 458-460.  Due the small amount of obtained plant material from micropropagation, the yield of the extracts was not sufficient to undertake additional experiments to achieve a better chemical profile of them. Moreover, this study revealed that an extended experiment could lead to better information regarding the supplementation.

  1. Materials & Methods: Why is there no control used in the study, where you do not add extra sucrose / charcoal? Or is this the T7 that is not explained? Or is the greenhouse experiment the control?

As explained in comment 7, the control is T1, with 3% of glucose (30 g/L). And the T7 treatment refers to the plants that grew in the greenhouse, therefore we have altered the text to replace T7 for GH for better clarity, considering germination in a greenhouse was just a parallel experiment to compare with the development of plants from the micropropagation. That’s why T7 (now GH) was not included in Tables 1-3. We added this information in line 523.

  1. Materials & Methods: Explain T7.

We added this information about treatment T7 (now GH) in line 496. Additionally, the following text was included at the end of section 4.2: . It is important to note that the cultivation in greenhouse was carried out solely to observe the seeds behavior submitted to two different circumstances.

  1. Supplementary materials: Explain in this file what the figures mean.

We added this information in Supplementary materials: “The chromatograms of the crude ethanolic extracts from each treatment (T1-GH) of bulbs, leaves and roots of Zephyranthes irwiniana is described below.  α-terpinyl-acetate (blue column), ethyl linoleate (green column), lycorine (yellow column) and clionasterol (red column). X axis: Absolute Intensity (x108). Y axis: Retention time (min).”

Round 2

Reviewer 3 Report

Comments and Suggestions for Authors

Thanks for addressing all comments.